# Speed Regulation and Optimization of Sensorless System of Permanent Magnet Synchronous Motor

**Yan Zhang, Huacai Lu *** , **Minghu Li and Xiang Liu**

Key Laboratory of Electric Drive and Control of Anhui Province, Anhui Polytechnic University,
Wuhu 241000, China; mytlvyan@163.com (Y.Z.); lmh980816@163.com (M.L.); liuxiang10221022@163.com (X.L.)
* Correspondence: luhuacai@163.com

**Abstract:** Aiming at the problems of speed overshoot, slow convergence and poor anti-interference in the control of permanent-magnet synchronous motors (PMSMs) without a position sensor, a pulse vibration high-frequency signal injection method for a permanent-magnet synchronous motor with an improved sliding mode control was designed. Firstly, the improved approach rate function is combined with the improved non-singular fast terminal sliding mode surface to design the non-singular fast terminal sliding mode controller (NFTSMC), which is used in the speed loop to improve the speed convergence ability and reduce its overshoot. Secondly, in order to eliminate the influence of the band-pass filter on the system bandwidth in the traditional high-frequency injection method, a pulse vibration high-frequency signal injection method that injects high-frequency voltage signals and synchronous current signals into the $\hat{d}$ axis of the estimated two-phase rotation coordinate system $\hat{d}\hat{q}$ and the $\alpha\beta$ axis of the two-phase stationary coordinate system $\alpha\beta$ was designed to estimate the motor position and speed to achieve sensorless control. Finally, the above control strategy was compared with the speed loop PI and the traditional sliding mode controller (SMC) of the speed loop, respectively. The simulation and experimental results show that whether it is a no-load variable speed or fixed speed loading, the above control strategy can effectively reduce the speed overshoot, accelerate the speed convergence and improve the load capacity of the system.

**Keywords:** permanent-magnet synchronous motor; sensorless; high-frequency signal injection method; sliding surface; non-singular fast terminal sliding mode controller



## 1. Introduction

Permanent-magnet synchronous motors (PMSMs) are widely used in manufacturing and electric drives due to their high efficiency and high-power density [1,2]. The rotor position and speed information of the motor have an irreplaceable role in the field-oriented control and coordinate transformation, and the rotor position information is usually obtained by the mechanical sensor. However, the mechanical sensor is greatly affected by the environment and easy to damage. Additionally, its reliability is not high and the installation cost is large. Therefore, sensorless control systems are widely studied, in which the rotor position information is obtained and the observer method [3,4] is usually used for medium and high speeds, which are obtained indirectly by back EMF. Literature [5] uses the saturation function instead of the switching function to design an improved sliding mode observer, which reduces the speed jitter and improves the system stability. The high-frequency injection method is usually used for a zero low speed [6,7]. Literature [8] injects the current into a two-phase stationary coordinate system to obtain angular information, eliminating the need for a band-pass filter and reducing its impact on system bandwidth. The high-frequency injection method designed by the literature [9] based on beatless predictive control converts the current tracking error in the cost function into a voltage tracking error, which is robust to the parameter.

However, in a PMSM sensorless vector control system, there are still problems of a speed overshoot, slow convergence speed and poor anti-interference ability. In order

to solve these problems, nonlinear control theory is used for speed current double loop control, such as a sliding mode control [10,11],, predictive control [12] and intelligent control [13]. Among these, the sliding mode control has been widely studied because it is insensitive to the parameters of the motor itself and has strong anti-interference ability. In order to improve the interference and jitter problems caused by the switching delay of the traditional sliding mode controller (SMC), literature [14] designs a nonlinear proportional-terminal sliding surface and a fast terminal sliding mode surface for the current loop and speed loop, respectively, which effectively improves the speed overshoot and anti-interference ability. In literature [15], respectively, a new approach rate algorithm is designed for the double loop of the brushless DC motor to replace the constant velocity approach rate, which has a faster approach process, a smaller sliding range during the stable operation, suppresses the speed and current jitter, and makes the system run more stably. Literature [16] improves the absolute power approximation rate algorithm and observes the rotor position information through the state observer to compensate the rotor position in time, which effectively suppresses the speed overshoot; however, the speed convergence is slow. In literature [17,18] the high-order sliding mode controller is used to control the rotor suspension position and the state and position input estimation of vehicle tires when they are rubbed on the ground, and the simulation results verify that the rotor position has better dynamic characteristics and the validity of tire friction state estimation and unknown input estimation, respectively, and jointly verify that the high-order sliding mode controller can effectively improve the anti-interference performance of the control system. Literature [19,20] combines the sliding mode and predictive control to design a new integral sliding mode surface for a nonlinear system with multiple inputs and multiple outputs, which effectively enhances the input tracking characteristics of the system and can better cope with external interference and input changes. Literature [21–24] combines the improved fast terminal sliding mode controller with the disturbance observer for PMSM speed regulation, and designs an adaptive sliding mode controller, which effectively improves the overshoot, convergence and anti-interference performance of the control system.

Based on the above research, a high-frequency injection method for a PMSM based on the double-loop sliding mode control is proposed for the speed regulation of the PMSM sensorless control system, firstly, in order to prevent the singular problem which is caused by the use of a fast terminal sliding surface when the state variable is zero. This paper designed a non-singular fast terminal sliding mode controller (NFTSMC) for speed loop speed regulation. The NFTSMC consists of an improved non-singular fast terminal sliding surface and an improved approach rate function. Secondly, in order to improve the q-current tracking ability, a $q$-axis current loop controller based on a super-twisting algorithm is designed, and the stability of the speed loop and current loop controller is verified by using Lyapunov stability theory. Finally, a pulse vibration high-frequency voltage signal injection method is designed to directly extract the rotor speed and position information from the stationary coordinate system, which eliminates the influence of the band-pass filter on the system bandwidth.

## 2. Double-Loop Sliding Mode Controller Design

The research object of this paper is a surface-mount PMSM, regardless of core saturation and eddy current hysteresis loss, and the vector control strategy of $i_d^* = 0$ is used to obtain the state equation of the PMSM under the two-phase rotating coordinate system $dq$ [25]:

$$\frac{di_q}{dt} = \frac{1}{L}(u_q - Ri_q - n_p\omega\varphi) \tag{1}$$

$$\frac{d\omega}{dt} = \frac{1}{J}(1.5n_p\varphi i_q - B_a\omega - T_L) \tag{2}$$

where $u_q$ and $i_q$ are the $q$-axis voltage and current, $R$ and $L$ are the stator resistor and inductor, respectively, $\omega$ is the mechanical angular velocity of the rotor, $\varphi$ is a magnetic link, $n_p$ is the number of pole logarithms which refer to the number of pairs of N and S poles of a permanent magnet rotor (or stator), $J$ is the moment of inertia, $B_a$ is the damping coefficient and $T_L$ is the load torque.

### 2.1. The Non-Singular Fast Terminal Sliding Mode Controller Design

In order to improve the problems of speed overshoot, slow convergence and weak anti-interference ability, the sliding mode controller is used instead of the PI controller in the speed loop, while the approach rate function of the traditional SMC is the switch symbol function, which can easily affect the stability of the system at the zero point. In addition, in order to accelerate the speed convergence, this paper combines the improved approach rate function with the non-singular fast terminal sliding mode surface to design the non-singular fast terminal sliding mode controller (NFTSMC).

We define the state variable $x$ for PMSM as:

$$\begin{cases} x_1 = \int \omega_{ref} - \omega \\ x_2 = \omega_{ref} - \omega \end{cases} \tag{3}$$

where $\omega_{ref}$ is the rotor reference speed.

In order to enhance the speed tracking and convergence performance, a non-singular fast terminal sliding surface is designed:

$$s = x_1 + \frac{1}{m}|x_1|^\alpha sign(x_1) + \frac{1}{n}|x_2|^{\beta/\gamma} sign(x_2) \tag{4}$$

where $\alpha > 1, m > 0, n > 0, \beta, \gamma \in N^+$ is a positive odd number and $1 < \beta/\gamma < 2$.

In order to reduce the speed jitter and enhance the anti-interference ability of the system, the design approach rate function is as Equation (5), the approach stage $\lambda(1 - \eta)sign(s)$ dominates the speed to quickly converge to $\omega_{ref}$, and the sliding stage $le^{|s|}s$ dominates, so that the speed jitter decreases.

$$\begin{cases} \dot{s} = -\frac{\beta}{n\gamma}\left|x_2\right|^{\beta/\gamma-1}[\lambda(1-\eta)sign(s) + le^{|s|}s], \ \lambda > 0, l > 0 \\ \eta = \begin{cases} e^{-e|s|} & |s| < 1 \\ 1 - e^{|s|} & |s| \geq 1 \end{cases} \end{cases} \tag{5}$$

From Equation (2) to Equation (5), the $q$-axis reference current $i_q^*$ output by the speed ring controller is:

$$i_q^* = \frac{2J}{3n_p\varphi}\left[\frac{\beta}{n\gamma}|x_2|^{2-\beta/\gamma}sign(x_2)(1 + \frac{\alpha}{m}|x_1|^{\alpha-1}sign(x_1)) + \lambda(1-\eta)sign(s) + le^{|s|}s + \frac{B_a}{J}\omega + \frac{T_L}{J}\right] \tag{6}$$

Stability analysis: we suppose the Lyapunov function to be $V = \frac{1}{2}s^2$ and derive its derivatives:

$$\dot{V} = s\dot{s} = s\left\{-\frac{\beta}{n\gamma}|x_2|^{\beta/\gamma-1}[\lambda(1-\eta)sign(s) + le^{|s|}s]\right\} = -\frac{\beta}{n\gamma}|x_2|^{\beta/\gamma-1}[\lambda(1-\eta)|s| + le^{|s|}s^2] \tag{7}$$

From the value range of each parameter, we can know $\dot{V} \leq 0$, so the designed speed loop controller is stable.

### 2.2. q-Axis Current Loop Controller Design

In order to improve the tracking performance of the $q$-axis current and enhance the anti-back EMF disturbance, a $q$-axis current controller based on the super-twisting algorithm [26] is designed, and its sliding mode surface is designed as:

$$s_q = i_q{}^* - i_q \tag{8}$$

We design the approach rate function as the super-twisting algorithm:

$$\dot{s}_q = -k_p|s_q|_r sign(s_q) - k_i sign(s_q) \tag{9}$$

where $k_p > 0$ and $k_i > 0$ are the proportional, integral coefficient, respectively, $r > 0$.

From Equations (1), (8) and (9), it can be seen that the output voltage of the $q$-axis current loop $u_q$:

$$u_q = L(Ri_q + n_p \varphi \omega + k_p|s_q|_r sign(s_q) + k_i sign(s_q)) \tag{10}$$

Stability analysis: Similarly, the Lyapunov function is designed as $V_q = \frac{1}{2}s_q{}^2$, and derives its derivatives:

$$\dot{V}_q = s_q \dot{s}_q = s_q(-k_p|s_q|^r sign(s_q) - k_i sign(s_q)) = -(k_p|s_q|^{r+1} + k_i|s_q|) \leq 0 \tag{11}$$

Therefore, the designed $q$-axis current loop is stable.

### 3. Design of Pulse Vibration High-Frequency Voltage Signal Injection Method

*3.1. Traditional Pulse Vibration High-Frequency Voltage Injection Method*

The start of the pulse vibration high-frequency voltage injection method mainly uses the non-ideal characteristics of the motor itself to estimate the speed and position signals of the motor. The main principle is that the high-frequency voltage signal injected on the stator side of the motor will form a high-frequency magnetic field in the motor. Because the motor has non-ideal characteristics, such as rotor structure convexity, rotor saturation convexity, etc., this non-ideal characteristic will modulate the high-frequency magnetic field, so that the stator side of the motor generates a current signal related to the position and speed of the motor, and obtains the speed and position information of the motor by extracting effective signals and demodulating.

The pulse high-frequency voltage injection method [27] only injects a high-frequency sinusoidal voltage signal in the $\hat{d}$ axis under the estimated two-phase rotation coordinate system $\hat{d}\hat{q}$, and obtains the rotor position observation value by extracting the amplitude information of the high-frequency current signal, and establishes the coordinate system shown in Figure 1 for estimating the rotor position.

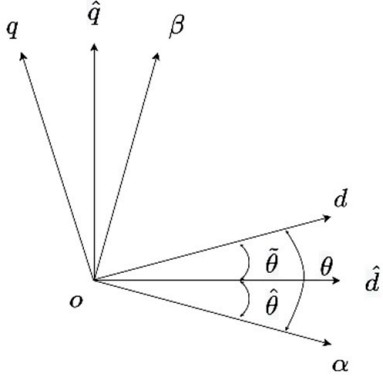

**Figure 1.** Position of the three axes.

In Figure 1, $\alpha\beta$ is the two-phase stationary coordinate system, $\theta$ and $\hat{\theta}$ are the actual position and estimated position of the rotor, and $\widetilde{\theta}$ is the difference between the two, namely:

$$\widetilde{\theta} = \theta - \hat{\theta} \tag{12}$$

With a high-frequency signal injection [20], the voltage equation for PMSM can be simplified to Equation (13),

$$
\begin{cases}
u_{dhin} = L_{dh} \dfrac{di_{dhin}}{dt} \\
u_{qhin} = L_{qh} \dfrac{di_{qhin}}{dt}
\end{cases}
\tag{13}
$$

where $u_{dqhin}$, $i_{dqhin}$ and $L_{dqh}$ are the high-frequency voltage, current, and inductance components of $dq$.

A high-frequency voltage signal as shown in Equation (14) is injected in the $\hat{d}\hat{q}$ coordinate system [28]:

$$
\begin{cases}
\hat{u}_{dhin} = U_{hin} \cos(\omega_{hin} t) \\
\hat{u}_{qhin} = 0
\end{cases}
\tag{14}
$$

where $\hat{u}_{dqhin}$ is the high-frequency voltage component of $\hat{d}\hat{q}$. $U_{hin}$ and $\omega_{hin}$ are injected high-frequency voltage amplitude and angular frequency, respectively.

Combined with the coordinate transformation of Equations (13) and (14) and Figure 1, the high-frequency response current of the $\hat{d}\hat{q}$ axis can be obtained as in Equation (15):

$$
\begin{cases}
\hat{i}_{dhin} = \dfrac{U_{hin}(L_{ave} + L_{sem}\cos(2\widetilde{\theta}))}{\omega_{hin}(L^2_{ave} - L^2_{sem})} \sin(\omega_{hin} t) \\
\hat{i}_{qhin} = \dfrac{U_{hin} L_{sem} \sin(2\widetilde{\theta})}{\omega_{hin}(L^2_{ave} - L^2_{sem})} \sin(\omega_{hin} t)
\end{cases}
\tag{15}
$$

where $\hat{i}_{dqhin}$ is the high-frequency current component of $\hat{d}\hat{q}$, $L_{ave} = \left(L_{qh} + L_{dh}\right)/2$ is the average inductance and $L_{sem} = \left(L_{qh} - L_{dh}\right)/2$ is a half-difference inductor.

The traditional pulse high-frequency voltage injection method passes $\hat{i}_{qhin}$ into the band-pass filter, then multiplies with the synchronous current signal $\sin(\omega_{hin} t)$ for amplitude modulation, and then passes through the low-pass filter to obtain $f\left(\widetilde{\theta}\right)$ containing the position error signal, that is:

$$
f(\widetilde{\theta}) = LPF(BPF(\hat{i}_{qhin}) \times \sin(\omega_{hin} t)) = \frac{U_{hin} L_{sem} \sin(2\widetilde{\theta})}{\omega_{hin}(L^2_{ave} - L^2_{sem})} = \frac{U_{hin}(L_{qh} - L_{dh})}{2\omega_{hin} L_{qh} L_{dh}} \sin(2\widetilde{\theta})
\tag{16}
$$

when $\hat{\theta}$ is very close to $\theta$ and $\sin(2\widetilde{\theta}) \approx 2\widetilde{\theta}$. Then $f\left(\widetilde{\theta}\right)$ is entered into the rotor position observer, and $f\left(\widetilde{\theta}\right)$ is first made to 0 through the PI regulator, and the estimated rotor speed ($\hat{\omega}$) can be obtained, and the rotor estimated angle ($\hat{\theta}$) can be obtained by integrating the estimated rotor speed, so as to realize the sensorless control of PMSM.

### 3.2. Improved Pulse High-Frequency Voltage Injection Method

The traditional pulse high-frequency voltage injection method cannot be adapted to motors without obvious salient poles, and the band-pass filter has a great influence on the system bandwidth. Therefore, in this paper, the two-phase stationary current is extracted directly from the two-phase stationary coordinate system, and the amplitude modulation is carried out with the synchronization signal after filtering by the low-pass filter, and the speed and position information is obtained by the phase-locked loop after the low-pass filtering again.

From Equation (13) and Figure 1, the current of the two-phase stationary coordinate system can be obtained, that is:

$$
\begin{bmatrix} \dfrac{di_{\alpha hin}}{dt} \\ \dfrac{di_{\beta hin}}{dt} \end{bmatrix} =
\begin{bmatrix} \dfrac{\cos\theta}{L_{dh}} & -\dfrac{\sin\theta}{L_{qh}} \\ \dfrac{\sin\theta}{L_{dh}} & \dfrac{\cos\theta}{L_{qh}} \end{bmatrix}
\begin{bmatrix} u_{dhin} \\ u_{qhin} \end{bmatrix}
\tag{17}
$$

where $i_{\alpha hin}$ and $i_{\beta hin}$ are currents in the two-phase stationary coordinate system when injected at high frequencies, and Equation (17) is obtained by turning it into the relationship between $\alpha\beta$ and $\hat{d}\hat{q}$ and combining Equation (14):

$$\begin{bmatrix} i_{\alpha hin} \\ i_{\beta hin} \end{bmatrix} = \begin{bmatrix} \frac{\cos\theta\cos\widetilde{\theta}}{L_{dh}} + \frac{\sin\theta\sin\widetilde{\theta}}{L_{qh}} \\ \frac{\sin\theta\cos\widetilde{\theta}}{L_{dh}} - \frac{\cos\theta\sin\widetilde{\theta}}{L_{qh}} \end{bmatrix} \frac{U_{hin}}{\omega_{hin}} \sin(\omega_{hin}t) \tag{18}$$

When $\widetilde{\theta}$ is small enough, Equation (18) can be reduced to:

$$\begin{bmatrix} i_{\alpha hin} \\ i_{\beta hin} \end{bmatrix} = \frac{U_{hin}\sin(\omega_{hin}t)}{\omega_{hin}L_{dh}} \begin{bmatrix} \cos\theta \\ \sin\theta \end{bmatrix} \tag{19}$$

Equation (19) is modulated with the synchronous current signal, and then the speed and position information can be obtained through the phase-locked loop, and the block diagram of the improved pulse high-frequency voltage injection method is shown in Figure 2.

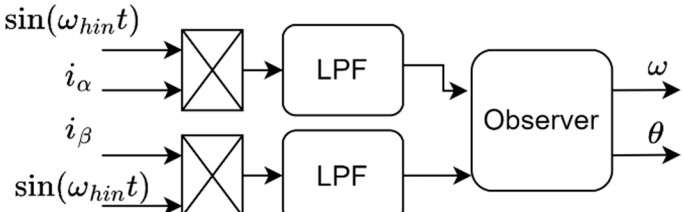

**Figure 2.** Improved pulse high-frequency voltage injection method.

The high-frequency voltage injected in this article has a frequency of 1000 Hz. The permanent magnet synchronous motor used in this article has a rated speed of 1000 r/min, the number of pole pairs is 4, and the sampling object is the phase current. In the rated case, the frequency of the phase current is the rated speed multiplied by the number of pole pairs divided by the power frequency. Because the frequency of China's power supply is 50 Hz, so in the rated case, the frequency of the phase current is 80 Hz, considering that the motor is overloaded, the instantaneous speed of the motor can exceed 1000 r/min, retain a certain margin and the cutoff frequency is taken as 150 Hz. Before the motor is run, only the DC bias is considered, so the cutoff frequency can be set very low, and the cutoff frequency is taken as 1 Hz, so the frequency bandwidth is 149 Hz.

## 4. Simulation Analysis

The block diagram of the system block diagram of the high-frequency injection method of the PMSM based on double-ring sliding mode control is shown in Figure 3, which is simulated and verified in MATLAB/Simulink, and compared with the system with the speed loop PI controller and the traditional SMC, the motor parameters used in this paper are from the AISim semi-physical simulation experimental platform. In the AISim semi-physical simulation system, the specifications of both the drive motor and the load motor are the same. The drive motor is controlled by a special servo control system and the load motor is controlled by a universal servo control system, which are connected with each other by using couplings on the same motor base. The load motor rotates under the drive of the drive motor, similar to a generator, and the energy generated needs to be released output. Currently, the energy is released by generating heat through the braking resistor. The brake resistor is a ripple resistor which has good heat dissipation performance. The 1.5 KW load motor is configured with a 1.5 KW, 40 ohm braking resistance, and the more detailed motor parameters are shown in Table 1.

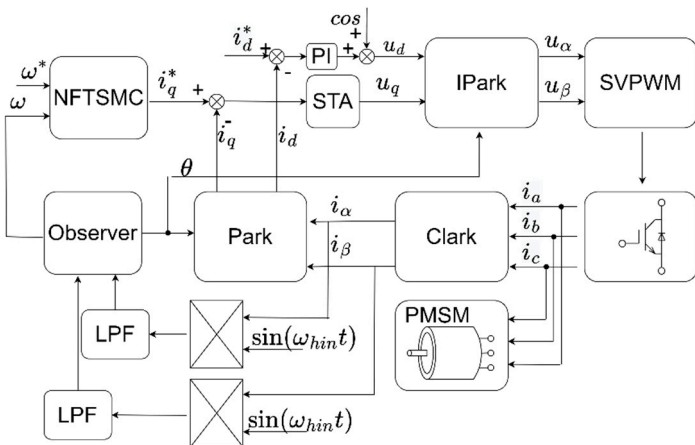

**Figure 3.** High-frequency injection method of PMSM controlled by double-loop sliding mode.

**Table 1.** Motor parameters.

| Parameters | Values | Parameters | Values |
|---|---|---|---|
| Rated power (kw) | 1.5 | Flux linkage $\varphi$ (Wb) | 0.1827 |
| Rated speed $\omega$ (rpm) | 1000 | Pole pairs $n_p$ | 4 |
| Back EMF (V·min/rad) | 138/1000 | Stator inductance L (H) | 0.00665 |
| Stator resistance R ($\Omega$) | 1.84 | Rotary inertia J (kg·m$^2$) | 0.00277 |
| Rated torque (N·m) | 15 | Instantaneous maximum torque (N·m) | 45 |
| Rated phase current (A) | 7.3 | Weight (kg) | 12.6 |
| Torque coefficient (N·m/A) | 2.05 | Encoder (P/R) | 2500 |

Figure 4 shows the speed response waveform of the three controllers under the condition of setting the reference speed $\omega_{ref}$ = 200 r/min and adding the load $T_L$ = 5 N at 0.05 s. It can be seen from Figure 4 that the speed loop PI controller converges to $\omega_{ref}$ after 20 ms after a no-load start, the maximum overshoot is 23 r/min, and after 10 ms with 5 N load, it converges again to $\omega_{ref}$, and the maximum overshoot is −12 r/min. The traditional SMC converges to $\omega_{ref}$ after 8.5 ms after a no-load start-up, overshoots 6 r/min, and converges to $\omega_{ref}$ after 2 ms after a 5 N load, with a maximum overshoot of −5 r/min. The improved non-singular fast terminal sliding mode controller (NFTSMC) converges to $\omega_{ref}$ after 6 ms after a no-load start-up, overshoot 1 r/min, and converges again to $\omega_{ref}$ after 1 ms with a 5 N load, with a maximum overshoot of −2 r/min. From the above analysis, it can be seen that the speed convergence performance and anti-interference performance of the NFTSMC are better than the PI and traditional SMC. The analysis above is summarized in Table 2.

**Table 2.** Overshoot of three controllers in different conditions.

| | Condition 1 | Condition 2 |
|---|---|---|
| Time (s) | 0 | 0.05 |
| Reference speed $\omega_{ref}$ (r/min) | 200 | 200 |
| Load $T_L$ (N) | 0 | 5 |
| Overshoot of PI (r/min) | 23 | −12 |
| Overshoot of SMC (r/min) | 6 | −5 |
| Overshoot of NFTSMC (r/min) | 1 | −2 |

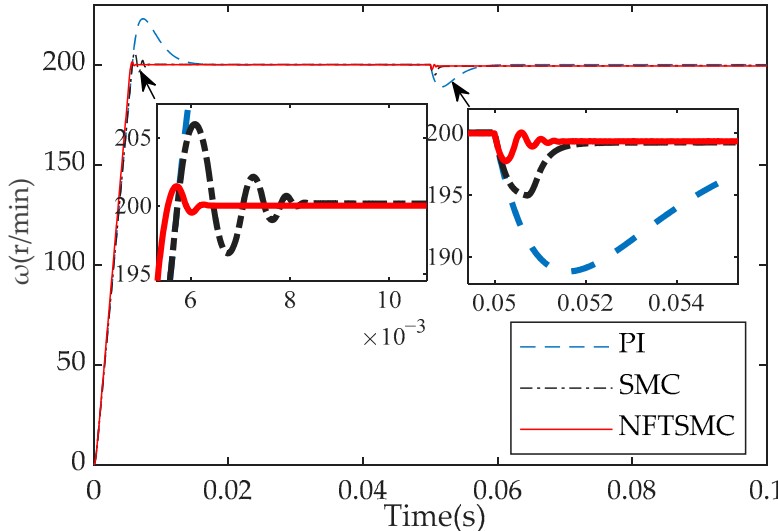

**Figure 4.** Three types of controllers are loaded at fixed speed.

Figure 5 shows the speed response of the three controllers under no-load conditions of the initial reference speed $\omega_{ref}$ = 10 r/min and set $\omega_{ref}$ = 300 r/min at 0.05 s. It can be seen from Figure 5 that the speed PI after starting converges to $\omega_{ref}$ = 10 r/min after 12 ms, overshoots 2 r/min, and converges to $\omega_{ref}$ = 300 r/min after 25 ms after a variable speed and overshoot 30 r/min. After the traditional SMC starts, it converges to $\omega_{ref}$ = 10 r/min after 1.4 ms, overshoots by 2 r/min, and the variable speed reaches a stable speed again after 11 ms, but it is 2.5 r/min higher than the given reference speed; that is, it is always overshot by 2.5 r/min. After NFTSMC starts, it converges to $\omega_{ref}$ = 10 r/min after 0.7 ms, without an overshoot, and after 8 ms after changing speed, it converges to $\omega_{ref}$ = 300 r/min and overshoot 1 r/min. From the above analysis, it can be seen that under the no-load variable speed condition, NFTSMC converges to a given speed faster than the PI and SMC and the overshoot is smaller. The analysis above is summarized in Table 3.

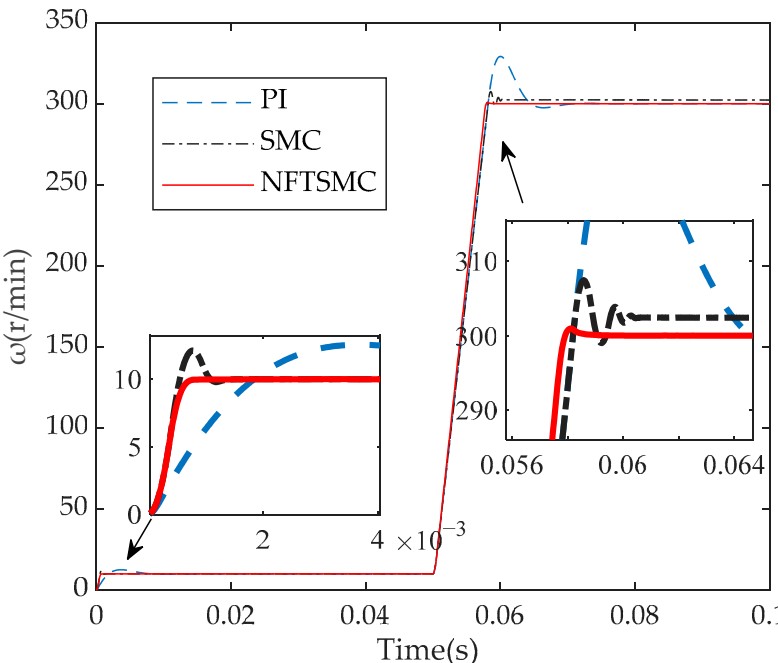

**Figure 5.** No-load variable speed of three controllers.

**Table 3.** Overshoot of three controllers at no-load.

|  | Condition 1 | Condition 2 |
|---|---|---|
| Time (s) | 0 | 0.05 |
| Reference speed $\omega_{ref}$ (r/min) | 10 | 300 |
| Overshoot of PI (r/min) | 2 | 30 |
| Overshoot of SMC (r/min) | 2 | 2.5 |
| Overshoot of NFTSMC (r/min) | 0 | 1 |

Figures 6 and 7 are $\omega_{ref}$ = 200 r/min, when the load ($T_L$ = 5 N) is added at 0.05 s, the actual speed and angle of the motor and the speed and angle estimated by the rotor position observer are compared with the comparison chart, where the small figure represents the difference between the estimated and the actual value. It can be seen from Figures 6 and 7 that the speed error is between −0.07–0 r/min and the angle tracking error is between −0.002–0 rad, indicating that the control strategy designed in this paper has good speed and angle tracking characteristics.

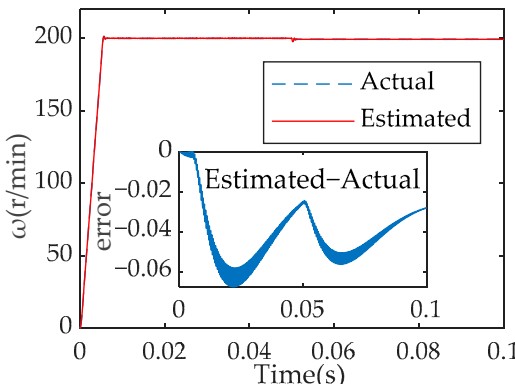

**Figure 6.** Speed tracking.

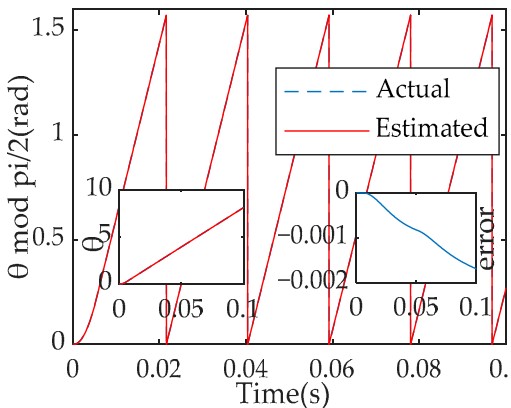

**Figure 7.** Angle tracking.

## 5. Experimental Verification

The experimental verification platform in this paper is an AISim semi-physical simulation platform, which is mainly composed of a host, real-time simulator, interface card and supporting equipment. The AISim simulation software (VxWorks 6.9) package is used to realize start-stop control, online monitoring, data post-processing and other operations for simulation running tests, as shown in Figure 8.

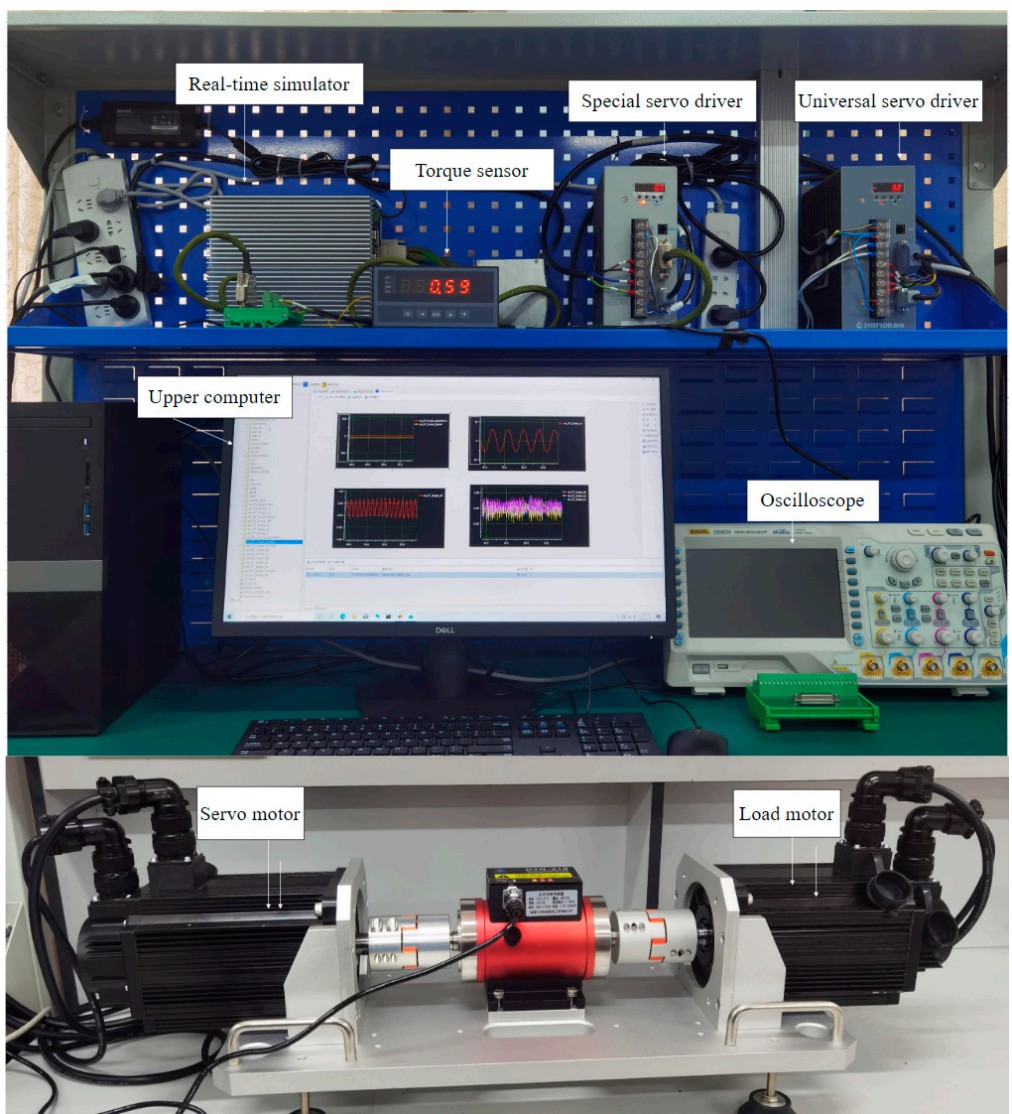

**Figure 8.** AISim semi-physical simulation platform.

Figure 9 shows the speed response of the three controllers when $\omega_{ref}$ is 50 r/min. The difference between the speed response of the three controllers (PI, SMC, NFTSMC) and the reference speed is captured in the small figure in Figure 9. As shown in Figure 9, the maximum overshoot of the three controllers (PI, SMC, NFTSMC) is 40 r/min, 20 r/min, and 0 r/min, respectively. After about 4 s, 3 s, and 1 s, they fluctuate around the given speed, and the speed oscillation of the NFTSMC is the smallest in a steady state.

Figure 10 shows the initial given $\omega_{ref}$ = 100 r/min, and sets $\omega_{ref}$ = 100 r/min when $t$ is 10 s, the speed response of the three controllers is as follows: in the initial state, the controller PI, traditional SMC, and NFTSMC overshoot 52 r/min, 30 r/min, and 0 r/min, respectively. After about 6 s, 3 s, and 1 s, they fluctuate around the given speed. As shown in Figure 10, after the speed changes, the NFTSMC converges to the given speed faster than the PI and traditional SMC, and the overshoot is smaller.

Figure 11 shows $\omega_{ref}$ = 100 r/min at the initial state. The 1 N load is added when $t$ is 10 s. After loading, the PI, traditional SMC, and NFTSMC of the three controllers' overshoot are −27 r/min, −22 r/min, and −13 r/min, respectively. Therefore, as shown in Figure 11, the NFTSMC has stronger load-bearing capacity than the PI and traditional SMC.

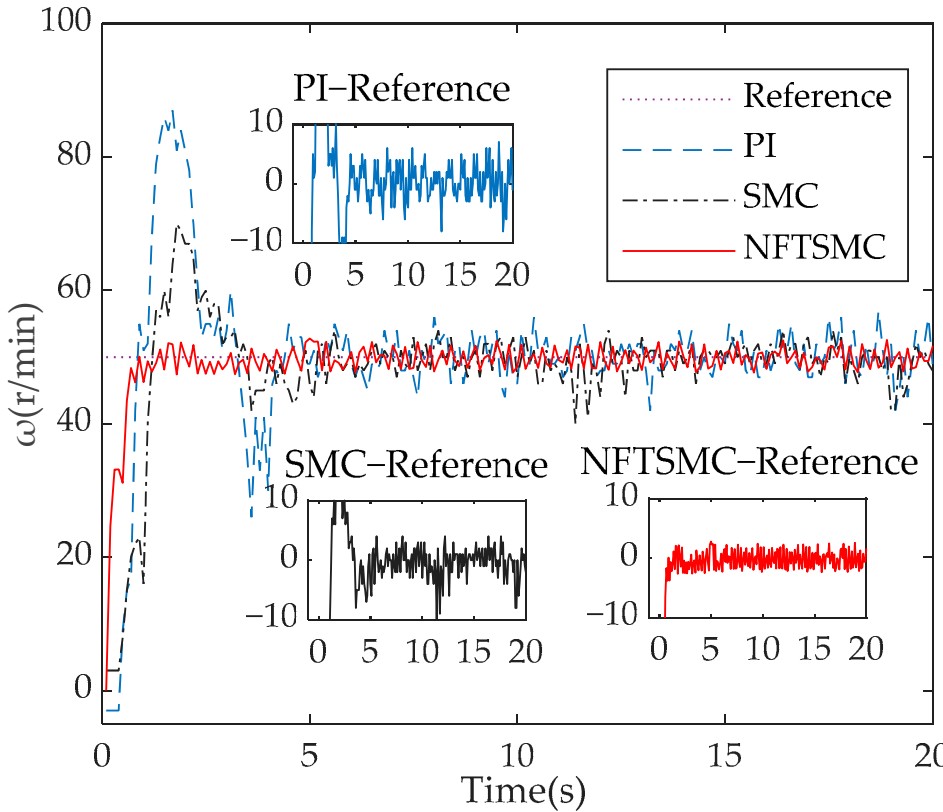

**Figure 9.** Comparison of three controller speed tracking.

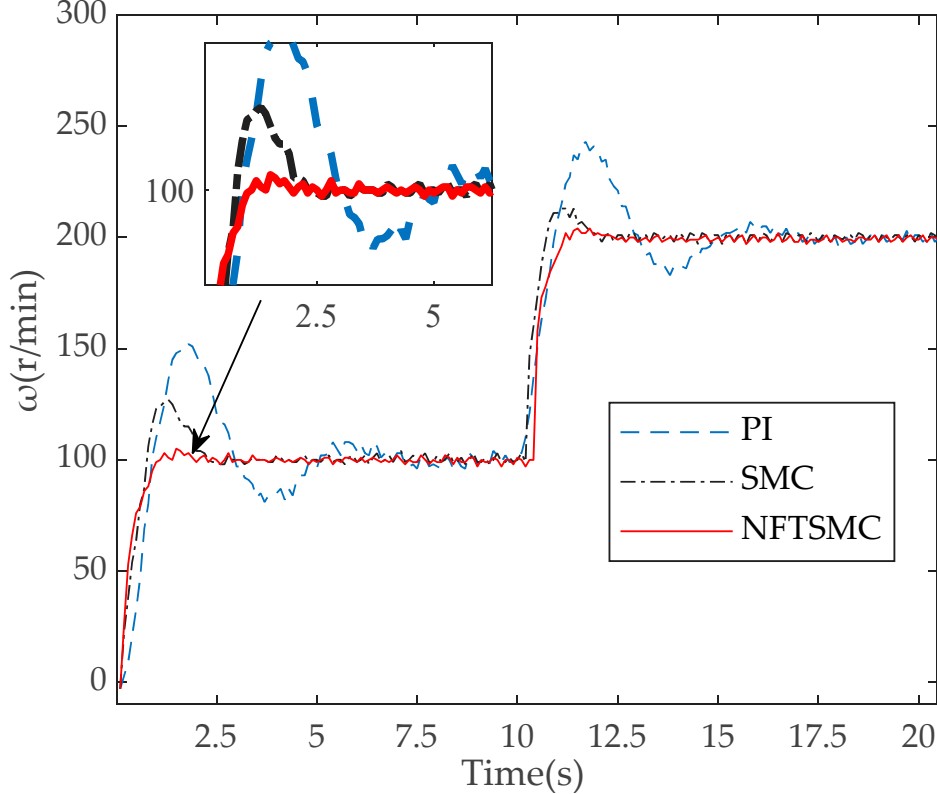

**Figure 10.** Three controller no-load variable speed.

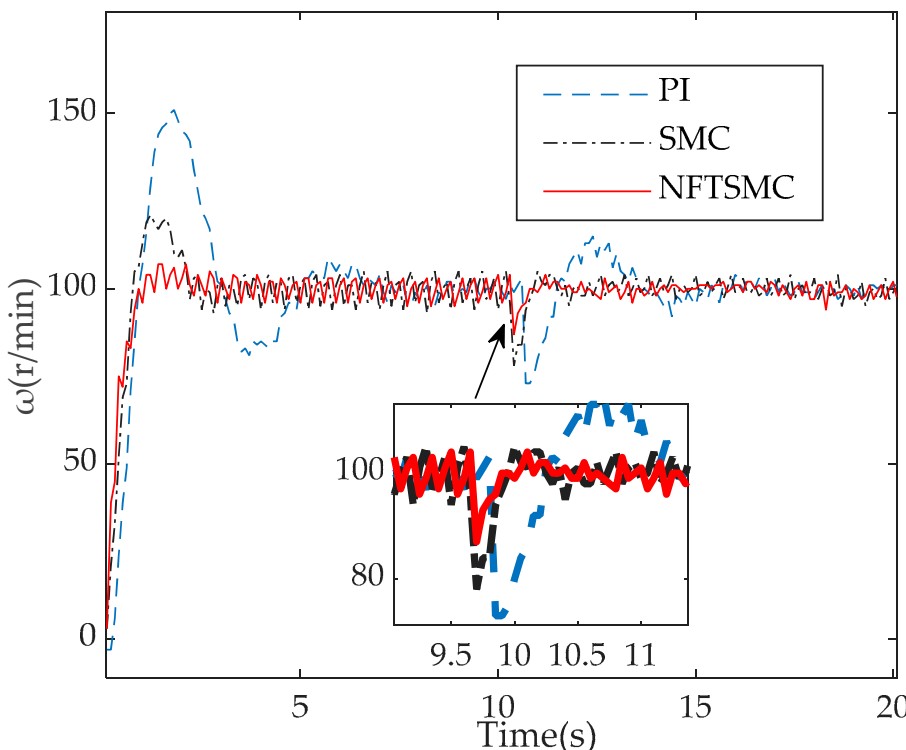

**Figure 11.** Three controllers are loaded at a fixed speed.

Figure 12 is given the reference speed $\omega_{ref} = 50\cos(0.5\pi t)$ r/min and load $T_L = 1$ N, the estimated speed of the rotor observer tracking given the reference speed effect diagram. Where Figure 12a shows the speed tracking effect when the speed loop is the PI, SMC and NFTSMC respectively, from Figure 12a it can be seen that the PI due to the use of the integrator caused by the speed tracking lag is serious, the tracking effect is very poor, and the error range of the PI tracking reference speed is −50–15 r/min. Therefore, the subsequent analysis no longer compares the speed loop PI controller, but only compares the speed loop SMC and NFTSMC used in this paper. Figure 12b is an enlarged view of the SMC and NFTSMC tracking reference speed curves, and when the steady state can be seen from Figure 12b, the error range of the SMC tracking reference speed is: −10–20 r/min, while the error range of the NFTSMC tracking reference speed is: −3–3 r/min; In summary, the tracking effect of the NFTSMC is much better than that of the PI and SMC when $\omega_{ref} = 50\cos(0.5\pi t)$. The analysis above is summarized in Table 4.

**Table 4.** Error range of three controllers at load.

|  | Condition |
| --- | --- |
| Time (s) | 0 |
| Reference speed $\omega_{ref}$ (r/min) | 50cos(0.5πt) |
| Load $T_L$ (N) | 1 |
| Error range of PI (r/min) | −50–15 |
| Error range of SMC (r/min) | −10–20 |
| Error range of NFTSMC (r/min) | −3–3 |

Figure 13 shows the SMC and NFTSMC tracking reference speed effect chart when the initial given reference speed $\omega_{ref} = 50\cos(0.5\pi t)$ r/min and $T_L = 1$ N load to make $T_L = 2$ N when t = 10 s. It can be seen from Figure 13 that after the load is increased, the maximum downward overshoot of SMC is 23 r/min, while the maximum downward overshoot of the NFTSMC is 2 r/min. It can be concluded that the load-carrying capacity of the NFTSMC is stronger than that of the SMC. The analysis above is summarized in Table 5.

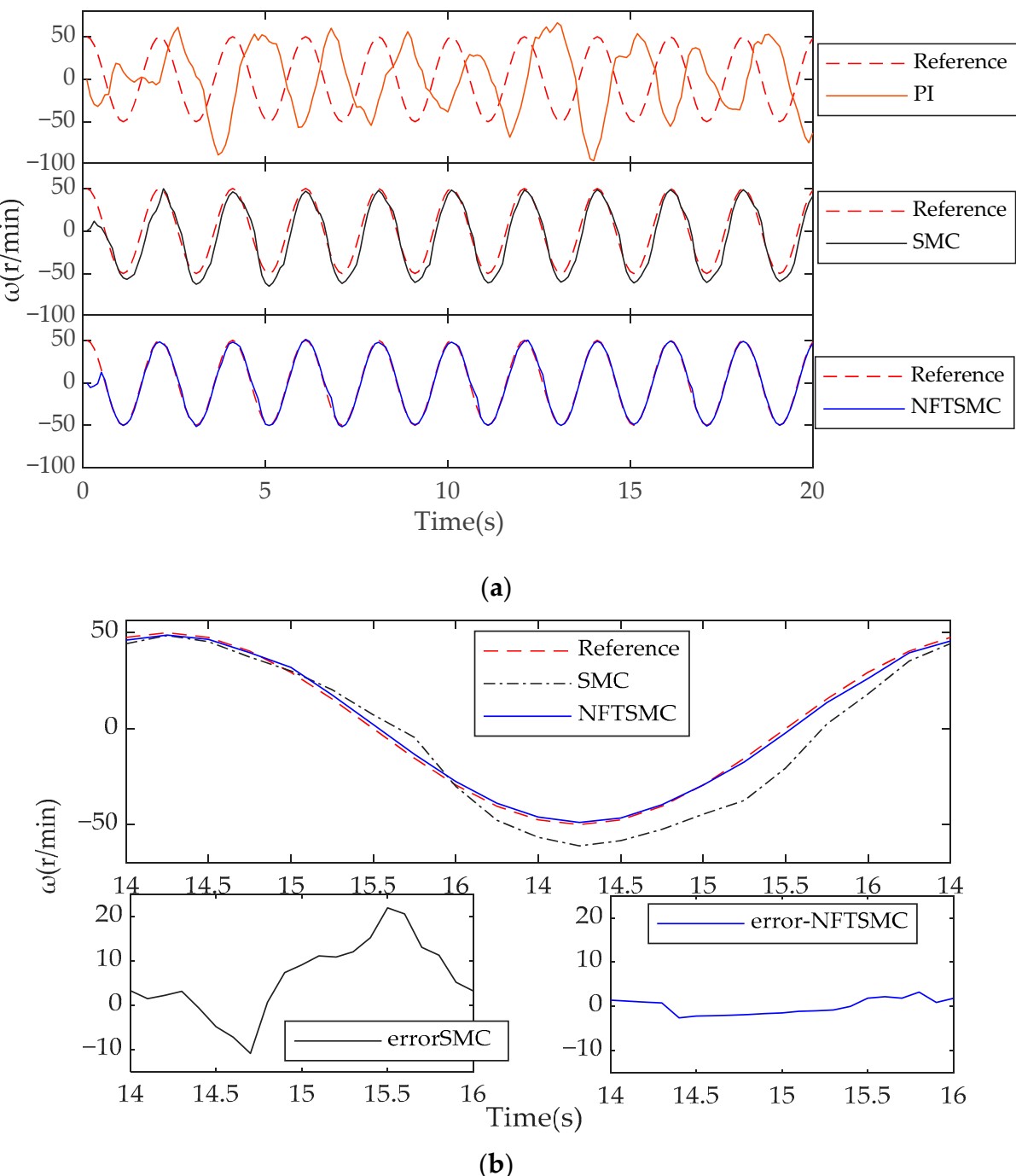

**Figure 12.** Two controllers with on-load variable speed. (**a**) Full speed range. (**b**) Partial zoom-in.

**Table 5.** Overshoot of two controllers at no-load.

|  | Condition 1 | Condition 2 |
| --- | --- | --- |
| Time (s) | 0 | 10 |
| Reference speed $\omega_{ref}$ (r/min) | 50cos(0.5πt) | 50cos(0.5πt) |
| Load $T_L$ (N) | 1 | 2 |
| Overshoot of SMC (r/min) | 12 | 23 |
| Overshoot of NFTSMC (r/min) | 1 | 2 |

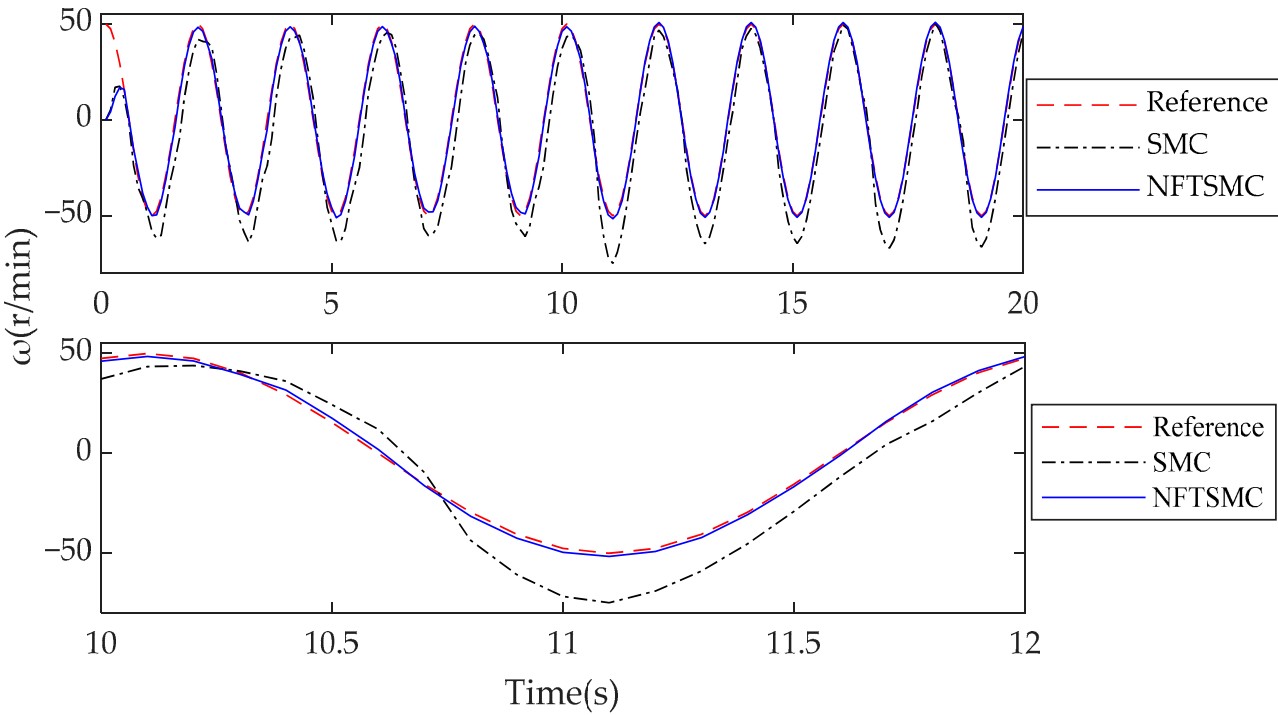

**Figure 13.** Two controllers are loaded at variable speed.

Figure 14 is a given reference speed $\omega_{ref} = 50 \cos(0.5\pi t)$ r/min, $T_L = 1$ N, the relationship between the reference speed, the actual speed of the motor and the estimated speed of the observer; the steady state can be seen from Figure 14, the estimated speed can better track the actual speed and the reference speed and the tracking error fluctuates between $-4$ and 4 r/min, occasionally reaching 12 r/min.

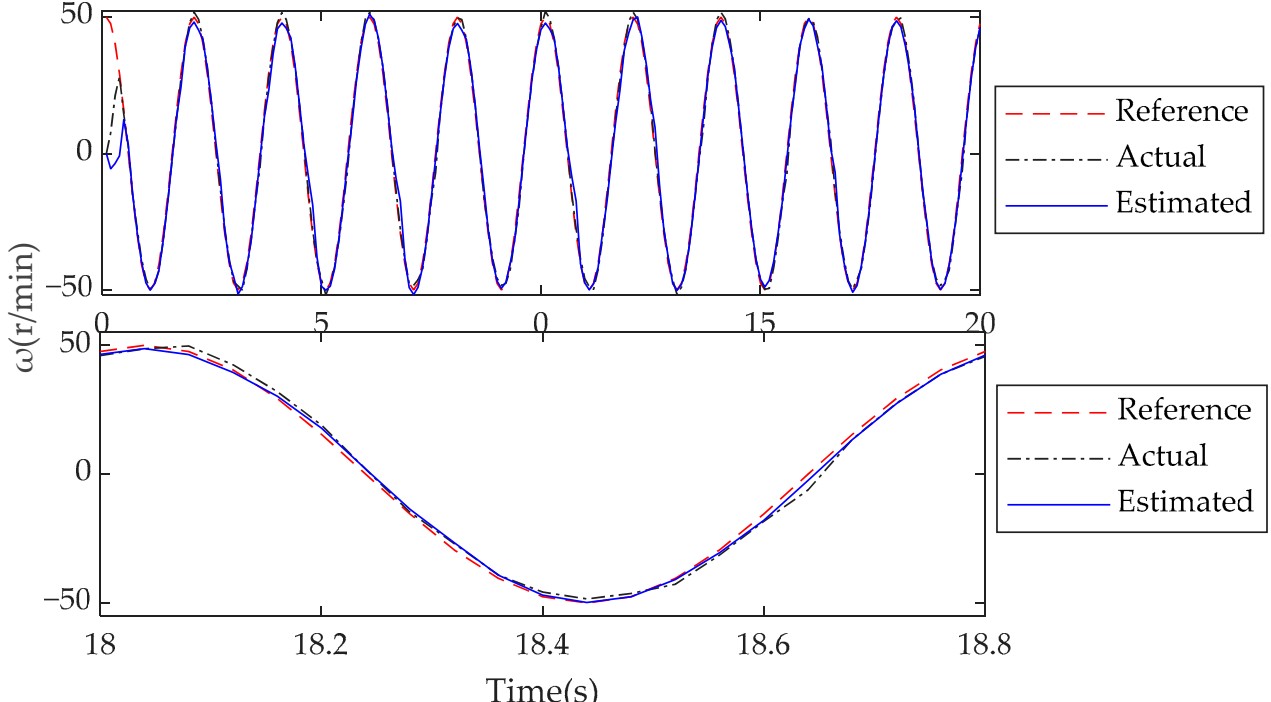

**Figure 14.** Estimated speed shifting tracks actual and reference speeds.

Figure 15 is given the reference speed $\omega_{ref}$ = 50 r/min, the observer estimates the speed tracking reference speed and the actual speed response curve of the motor; it can be seen from Figure 15 that the maximum tracking error between the estimated speed and the actual speed at startup is 20 r/min, and after stabilization, the tracking error fluctuates between −3 and 3 r/min, indicating that the estimated speed can effectively track the actual speed of the motor.

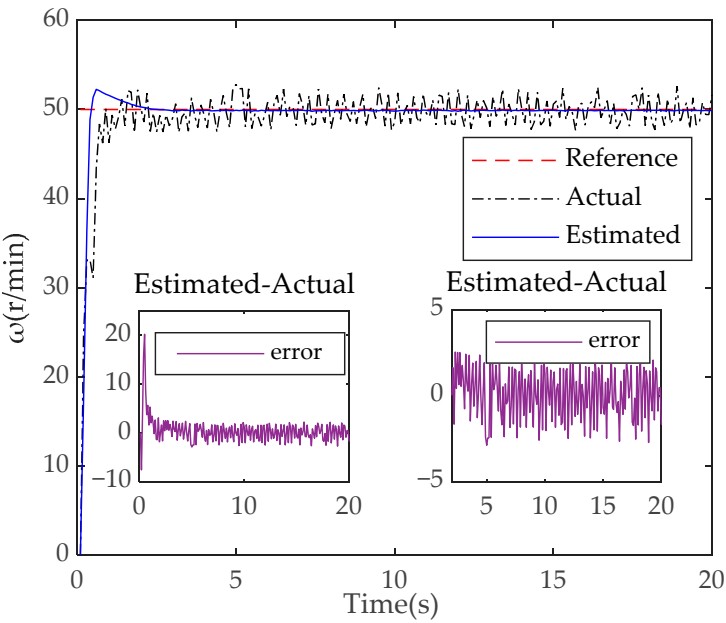

**Figure 15.** Estimated speed control tracks actual and reference speeds.

### 6. Conclusions

In order to improve the speed overshoot, slow convergence and bad load in the sensorless control system of the PMSM, in this paper, a compound control strategy based on a double-loop sliding mode control was proposed for the PMSM pulse vibration high frequency voltage injection method. The system was improved from the following three aspects:

1.  Design of NFTSMC. A non-singular fast terminal sliding mode controller (NFTSMC) based on an improved non-singular fast terminal sliding mode surface and improved approach rate function was designed to reduce the speed overshoot and accelerate the speed convergence.
2.  A $q$-axis current loop controller based on a super-twisting algorithm was designed to improve the $q$-axis current tracking effect and make the speed tracking smoother.
3.  To remove the bandpass filter, the pulse vibration high-frequency signal injection method that injects a high-frequency voltage signal and synchronous current signal into the $\hat{d}$ axis of the estimated two-phase rotation coordinate system $\hat{d}\hat{q}$ and the $\alpha\beta$ axis of the two-phase stationary coordinate system $\alpha\beta$ was designed to estimate the motor position and speed to achieve sensorless control.

According to the above simulation and experimental analysis, the proposed control strategy can effectively reduce the speed overshoot and convergence time, and improve the anti-interference ability of the system.

**Author Contributions:** Conceptualization, Y.Z.; methodology, M.L. and X.L.; software, Y.Z.; validation, H.L., Y.Z., M.L. and X.L.; formal analysis, H.L.; investigation, M.L. and X.L.; resources, H.L.; data curation, Y.Z., M.L. and X.L; writing—original draft preparation, Y.Z.; writing—review and editing, H.L., and Y.Z.; visualization, M.L. and X.L.; supervision, H.L.; project administration, H.L.; funding acquisition, H.L. All authors have read and agreed to the published version of the manuscript.

**Funding:** This work is supported by Anhui Provincial Natural Science Foundation (No. 1908085QE247), and the Open Research Fund of AnHui Key Laboratory of Detection Technology and Energy Saving Devices (No. JCKJ2022A04).

**Data Availability Statement:** The processed data required to reproduce these findings cannot be shared at this time as the data also form part of an ongoing study.

**Conflicts of Interest:** The authors declare no conflict of interest.

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
