# Peer review of "Speed Regulation and Optimization of Sensorless System of Permanent Magnet Synchronous Motor"

_machines, doi:10.3390/machines11060656_

Round 1

Reviewer 1 Report

a pulse vibration high-frequency signal injection method for permanent magnet synchronous motor with improved sliding mode control was designed in this article, and then the speed overshoot, slow convergence and poor anti-interference could be improved. In addition, the comparison to the normal PI model is conducted, and results show that the speed overshoot is reduced and the anti-interference ability is improved. The detailed comments are listed as follows

1, the critical parameters used in the simulation and experiment should be presented in a Table, and the servo motor used in the experiment should be introduced.

2, how to choose the frequency bandwidth used in the high-frequency voltage injection model? Please give more details!

3, in the title, the speed regulation of the PMSM without sensor is mentioned, how to realize the startup of the PMSM without sensors?

4, some figures should be revised, for example, the legend in Figure 12 (b) should be located outside of the data curve.

5, how to tackle the situation when the Lave is equal to Lsem in equation (15) if the q-axis inductance is same as the d-axis inductance.?

Moderate editing of English language required

Author Response

Dear Reviewer

        Thank you for the precious time you devoted to the paper. Please see the attachment.

Sincerely,

Yan Zhang

Reviewer 2 Report

The paper presents a Speed regulation of sensorless system of permanent magnet synchronous motor.

This interesting and well written paper should be published after minor improvements.

1.     The introduction situates the research work with many scientific references. However, the numbering of the references should be in increasing order. The second sentence of the last paragraph (Firstly, in order to prevent ….) is a bit long and difficult to understand. The authors should improve the redaction.

2.     In section 2, the authors should remind the concept of “number of pole logarithms” for helping the reader who has not read [25].

3.     The subsection titles should avoid acronyms.

4.     At the end of section 3.2, the authors should give an example of the frequency of the high frequency voltage and the low pass filter frequency. 

5.     In figure 6, the presentation of the small subfigures should be improved. I understood that the error is the absolute error in rpm; is it the case?

6.     In figure 7, I understood that the two curves are superimposed. But I cannot understand the first subfigure.

7.     The proportions of the picture in figure 8 seems strange. This photograph appears to have been expanded horizontally.

8.     In the conclusion, the double numbering (1. 2. 3.) should be removed.

Author Response

Dear Reviewer 

       Thank you for the precious  time you devoted to the paper. Please see the attachment.

Sincerely

Yan Zhang
